# The Roles of Liver Fibrosis Scores and Modified Stress Hyperglycemia Ratio Values in Predicting Contrast-Induced Nephropathy after Elective Endovascular Infrarenal Abdominal Aortic Aneurysm Repair

**DOI:** 10.3390/healthcare11060866

**Published:** 2023-03-16

**Authors:** Orhan Guvenc, Mesut Engin, Filiz Ata, Senol Yavuz

**Affiliations:** 1Medical Faculty, Departments of Cardiovascular Surgery, Uludağ University, Bursa 16310, Turkey; 2Department of Cardiovascular Surgery, Bursa Yuksek Ihtisas Training and Research Hospital, University of Health Sciences, Bursa 16310, Turkey; 3Department of Anesthesiology and Reanimation, Bursa Yuksek Ihtisas Training and Research Hospital, University of Health Sciences, Bursa 16310, Turkey

**Keywords:** aortic aneurysm, endovascular procedure, liver fibrosis, hyperglycemia, renal injury

## Abstract

Endovascular aortic repair (EVAR) methods are higher preferred for the treatment of patients with abdominal aortic aneurysms (AAAs). Various markers, including the neutrophil-lymphocyte ratio, have been used to predict the risk of contrast-induced nephropathy (CIN). In this study, we aimed to investigate the role of fibrosis-4 score (FIB-4), aspartate transaminase to platelet ratio index (APRI), and modified stress hyperglycemia ratio (mSHR) values in predicting CIN. Patients who had undergone elective endovascular infrarenal abdominal aortic aneurysm repair in our clinic between January 2015 and January 2022 were included in this retrospective study. Patients who did not develop contrast-induced nephropathy after the procedure were identified as Group 1, and those who did were referred to as Group 2. A total of 276 patients were included in the study. The two groups were similar in terms of age, gender, body mass index, hypertension, diabetes mellitus, chronic obstructive pulmonary disease, hyperlipidemia, and left ventricular ejection fraction. In Group 2, the FIB-4 score, APRI, and mSHR values were significantly higher (*p* = 0.008, *p* < 0.001, and *p* < 0.001, respectively). In Group 2, the contrast volume and number of packed blood products used (median 1 (1–4) vs. 2 (1–5)) were significantly higher (*p* = 0.003 and *p* = 0.012, respectively). In this study, we demonstrated that we may predict the risk of CIN development with preoperatively calculated, noninvasive liver fibrosis scores and mSHR.

## 1. Introduction

Abdominal aortic aneurysms (AAAs) are a prominent cardiovascular disease, and patients with AAAs should be promptly diagnosed and treated. The mortality rate increases greatly when rupture develops, and in recent years, endovascular methods have become highly preferred in the treatment of these patients. These interventions are referred to as endovascular aortic repair (EVAR) [1]. However, there is a risk of renal injury due to the use of contrast media after endovascular interventions, and various inflammatory biomarkers have been the subject of scientific studies seeking to predict this risk. Various precautions, such as ensuring adequate hydration and paying attention to the amount of contrast, are recommended before the procedure. Various markers, including the neutrophil-lymphocyte ratio, have been used to predict the risk of contrast-induced nephropathy (CIN) [2].

Some noninvasive fibrosis scores showing fatty liver have been shown to be important parameters in the prognosis of cardiovascular diseases; the fibrosis-4 score (FIB-4) and aspartate transaminase to platelet ratio index (APRI) are among these scoring systems. Studies have shown their relationship with atherosclerosis. Song et al. analyzed 665 nonalcoholic fatty liver disease subjects without chronic liver disease or heart disease. They demonstrated a significant relationship between FIB-4 scores and coronary atherosclerosis in patients with nonalcoholic fatty liver disease [3]. In another study that included 3433 patients aged ≥40 years, the authors showed a correlation between atherosclerosis and liver fibrosis scores [4]. Moreover, in a study conducted on the general population, a high FIB-4 value was shown to be a risk factor for renal insufficiency [5].

It has recently been shown that stress-induced hyperglycemia affects the prognosis of cardiovascular diseases, independent of whether the patient has diabetes mellitus (DM) [6]. Moreover, in current studies, the stress hyperglycemia ratio (SHR), obtained from the reference glucose value and estimated average chronic glucose levels, has been an important prognostic marker [7].

In this study, we aimed to investigate the role of liver fibrosis scores obtained from preoperative routine blood values and modified stress hyperglycemia ratio (mSHR) values, which we adapted to our study, in predicting contrast-induced nephropathy after endovascular abdominal aortic aneurysm repair.

## 2. Materials and Methods

Patients who had undergone elective endovascular infrarenal abdominal aortic aneurysm repair in our clinic between January 2015 and January 2022 were included in the study. Patients who had undergone emergency treatment, patients with pre-procedural creatinine values above 2 mg/dL, those who received nephrotoxic treatment in the preceding week (antibiotherapy due to infection, chemotherapy, etc.), and those with missing data were excluded from the study. After applying the exclusion criteria, 276 consecutive patients were included in the study (Figure 1).

Preoperative hemograms of the patients (white blood cell, platelet, neutrophil, and lymphocyte counts and blood glucose value at the beginning of the procedure), biochemistry (alanine transaminase (ALT), aspartate aminotransferase (AST), urea, creatinine, C-reactive protein, sodium, potassium, and magnesium), intraoperative parameters (amount of contrast agent used, blood glucose value at the beginning of the procedure, and duration of the procedure), and postoperative data (daily creatinine values, blood glucose value at the end of the procedure) were all recorded. Patients who did not develop contrast-induced nephropathy after the procedure were identified as Group 1, and those who did were referred to as Group 2. Contrast-induced nephropathy was defined as “0.5 mg/dL or 25% or more increase in serum creatinine value within 48 h after use of contrast material” [8].

Calculation of Scores Used in the Study
FIB-4 score = age [years] × AST [IU/L]/(platelets [×10^9^/L] × ALT [IU/L] 1/2)
APRI = AST concentration (IU/L)/upper limit of normal AST (IU/L) × 100/platelet count (10^9^/L)
mABG = (Blood glucose level at the start of the procedure + blood glucose level at half an hour after the start of the procedure + blood glucose level at admission to the intensive care unit after the procedure)/3
mSHR = (mABG)/[(28.7 × glycosylated hemoglobin %) − 46.7].

### 2.1. Preoperative Variables

Hypertension was defined as the use of at least one antihypertensive drug and/or arterial blood pressure above 140/90 mmHg; hyperlipidemia as the use of antilipidemic therapy and/or blood low-density lipoprotein levels above 150 mg/dL; diabetes mellitus as antidiabetic medication use or fasting blood glucose level above 126 mg/dL or above 200 mg/dL during routine examinations; preoperative chronic obstructive pulmonary disease as a post-bronchodilator forced expiratory volume in 1 sec/forced vital capacity of <70%.

### 2.2. Endovascular Procedure

All patients whose preoperative preparations were completed were processed in the angiography laboratory. Invasive arterial monitoring was provided to all patients. Two preferably 18 G peripheral vein catheters were placed, and bladder catheterization was performed. Depending on the clinical condition of the patients, the surgical procedure was performed with general or local sedation. At the intervention site, both main femoral arteries were turned and suspended. A 7 French sheet was placed in the middle of the purse sutures placed on the artery. Heparin (10,000 units) was administered to the patients [9]. The diameter of the aorta, wall calcification, presence of thrombus in the vessel lumen, and length of the aneurysm in which the stent graft was to be placed were calculated. After positioning was radiologically confirmed using contrast-enhanced acquisition, the stent-graft was placed (Talent^®^ (Medtronic Vascular, Santa Rosa, CA, USA) or Endurant (Medtronic Vascular, Santa Rosa, CA, USA)). The guide wires were removed, and the purse stitches were tied. The surgical areas were closed, and patients were taken to our intensive care unit for close follow-up.

### 2.3. Statistical Analysis

For analysis, IBM SPSS Statistics for Windows version 21.0 (IBM Corp., Armonk, NY, USA) was used. The mean, SD, median (min–max), number, and frequency of the variables were used to express them. To analyze the normality of the numerical data, the Kolmogorov–Smirnov and Shapiro–Wilk tests were utilized. The Mann–Whitney U test was used to evaluate non-normally distributed variables, whereas Student’s t-test was used to study variables with a normal distribution. Categorical variables were compared using the chi-square test. Multivariate logistic regression analysis was used to examine the predictors of CIN. Values with a *p*-value of <0.2 in univariate analyses were used in the multivariate analysis models. Age, APRI, mSHR, preoperative creatinine value, contrast volume, and packed blood products used were included in Model 1. FIB-4 score, mSHR value, preoperative creatinine value, contrast volume, and packed blood products used were included in Model 2. The receiver operating characteristic (ROC) curve was applied to predict the effects of the FIB-4 score, APRI, and mSHR values. Statistical significance was defined as a *p*-value of 0.05 or lower.

## 3. Results

A total of 276 patients were included in the study. The median ages of the 221 patients in Group 1 and the 55 patients in Group 2 were 71 (65–88) years and 74 (65–82) years, respectively (*p* = 0.158). The two groups were similar in terms of gender, body mass index, hypertension, diabetes mellitus, chronic obstructive pulmonary disease, hyperlipidemia, and left ventricular ejection fraction. In addition, there was no difference between the two groups in terms of current medical treatment (angiotensin-converting enzyme inhibitor, angiotensin receptor blocker, acetylsalicylic acid, statin, and calcium channel blocker) (Table 1).

The patients’ preoperative laboratory values are presented in Table 2. There were no significant differences between the groups in terms of white blood cell count, hemoglobin, platelet count, blood urea nitrogen, aspartate aminotransferase, alanine aminotransferase, C-reactive protein, HbA1c, sodium, potassium, and magnesium levels. In Group 2, the mABG, FIB-4 score, APRI, and mSHR values were significantly higher (*p* = 0.008, *p* < 0.001, *p* < 0.001, and *p* < 0.001, respectively) (Table 2).

The operative and postoperative features of the patients are presented in Table 3. The two groups were similar in terms of operation times, type of anesthesia, and mortality rates. In Group 2, the contrast volume and packed blood products used were significantly higher (*p* = 0.003 and *p* = 0.012, respectively).

Multivariate logistic regression analysis was performed to predict the factors affecting the development of CIN after the EVAR procedure (Table 4). In the multivariate analysis Model 1, APRI (odds ratio (OR): 0.791, 95% confidence interval (CI): 0.495–0.894, *p* = 0.014), mSHR (OR: 1.462, 95% CI: 1.110–2.125, *p* < 0.001), pre-creatinine (OR: 3.935, 95% CI: 2.571–5.150, *p* < 0.001), and contrast volume (OR: 0.632, 95% CI: 0.398–0.814, *p* = 0.037) were determined as independent predictors of CIN. In Model 2, FIB-4 score (OR: 1.124, 95% CI: 1.060–1.592, *p* = 0.005), mSHR (OR: 1.639, 95% CI: 1.350–2.780, *p* = 0.002), pre-creatinine (OR: 4.120, 95% CI: 2.872–6.190, *p* < 0.001), and contrast volume (OR: 0.590, 95% CI: 0.386–0.70, *p* < 0.001) were determined to be independent predictors of CIN.

ROC curve analysis revealed that the cut-off value for mSHR was 1.49 (AUC: 0.801, 95% CI: 0.727–0.874, *p* < 0.001, with 70.9% sensitivity and 68.3% specificity), for FIB-4 score was 1.52 (AUC: 0.707, 95% CI: 0.629–0.785, *p* < 0.001, with 69.1% sensitivity and 64.8% specificity), and for APRI was 0.28 (AUC: 0.681, 95% CI: 0.596–0.766, *p* < 0.001, with 61.8% sensitivity and 66.7% specificity) (Figure 2).

## 4. Discussion

Today, the EVAR procedure is easily applied with a high success rate, particularly in infrarenal AAA treatments. However, CIN that occurs due to the use of contrast material can increase the mortality and morbidity of these procedures. For this reason, studies investigating CIN risk factors have been and are being conducted. In this clinical study, we showed for the first time that noninvasive liver fibrosis scores and mSHR values may be risk factors for the development of postoperative CIN.

FIB-4 scores and APRI values are important noninvasive parameters indicating liver fibrosis. In addition to being easily available, various studies have demonstrated their prognostic value in cardiovascular diseases. A study by Jin et al. included 5143 patients with stable coronary artery disease, demonstrated by coronary angiography, who were followed for a median of seven years. The FIB-4 score was associated with the severity of coronary artery disease. In addition, the risk of developing cardiovascular events was found to be significantly higher during follow-up when there was a high FIB-4 score (hazard ratio: 1.128, *p* = 0.012) [10]. In another study conducted by Chen et al., 3263 coronary artery disease patients were included, and the patients were followed for an average of 7.56 (interquartile range: 6.86–8.31) years. A total of 538 deaths occurred during this period, of which 319 were attributed to cardiovascular causes. In this study, FIB-4 scores and APRI values, which we also examined in our study, were found to be associated with cardiovascular and all-cause mortality [11].

Multiple variables involved in CIN cause the renal medulla to become hypoxic. Contrast medium inhibits mitochondrial enzyme activities, which leads to a rise in adenosine through the hydrolysis of adenosine triphosphate and a reduction in the activity of nitric oxide (NO) synthase. Reactive oxygen species are produced during adenosine catabolism, and they consume NO released from endothelial cells subjected to contrast media, along with endothelin and prostaglandin [12]. Additionally, alterations in the proximal tubular cells, podocytes, and mesangial cells’ structural and functional characteristics are linked to ectopic lipids. There is mounting evidence that, in the case of proteinuria, intracellular lipid deposition causes proximal tubule cell dysfunction [13].

The relationship between these parameters (FIB-4 score and APRI value), which noninvasively indicate liver fibrosis, and renal function has been demonstrated. In a study by Schleicher et al. involving a large cohort of the general population, patients were divided into two groups: those with a FIB-4 score  ≥  1.3 (n = 66,084) and those with a FIB-4 score <  1.3 (n = 66,084). The study group was followed for 10 years, and renal insufficiency status was recorded. In the patient group with high FIB-4 scores, renal failure developed significantly more often during the follow-up period. At the end of their study, the authors emphasized that the FIB-4 score may be a risk factor not only for liver-related risks but also for renal problems [5]. In another study, Seko et al. included 344 biopsy-proven nonalcoholic fatty liver disease patients. They aimed to investigate the risk factors for the progression of chronic kidney disease in these patients. As a result of this study, high FIB-4 score values were found to be associated with the development of chronic renal failure in DM patients [14]. In a recent study, the relationship between the development of CIN after elective percutaneous coronary interventions and liver fibrosis parameters was investigated. In this retrospective study, which included 5627 patients, CIN developed in 353 (6.3%) patients. The preoperatively calculated FIB-4 scores and APRI values were shown to be significantly associated with the development of CIN [15]. We also found a significant relationship in our study between high FIB-4 scores and APRI values and the development of CIN in patients who had undergone elective EVAR procedures.

Hyperglycemia occurs in acute clinical events. The hyperglycemia response of patients to acute conditions, independent of DM, affects clinical outcomes [16]. As a result, oxidative stress increases, prothrombotic pathways are activated, and endothelial damage occurs due to this sudden increase in glucose in the blood [17]. In one study, the role of ABG values in predicting the development of CIN after percutaneous coronary interventions in patients with non-ST elevation myocardial infarction was investigated. In the retrospective study, which included 281 patients, high ABG values were found to be associated with the development of CIN after the procedure [18]. In a meta-analysis investigation that included eight studies, it was shown that procedural hyperglycemia increased the risk of developing CIN after coronary angiography, regardless of whether the patient had DM [19].

In a similar direction, in recent studies, the SHR value was found to be more accurate than the ABG value in predicting mortality and morbidity in acute cardiovascular events [20]. In another study, the SHR value was found to be associated with early mortality after acute myocardial infarction [21]. In a study conducted by Chen et al., the effect of the SHR value on poor outcomes after mechanical thrombectomy was investigated in patients with ischemic strokes. One hundred and sixty patients were included in the retrospective study. At the end of the study, the authors determined that the SHR value calculated from the admission blood values was associated with poor outcomes in the first three months after the procedure [22]. Additionally, Yang et al. investigated the outcome-predicting value of SHR in coronary artery disease patients who had undergone percutaneous coronary interventions. A total of 4362 patients were enrolled in this retrospective observational study. At the end of their study, the authors determined that a high SHR value was associated with poor outcomes [23].

In addition, studies showing the relationship between renal injury and SHR have been reported in the literature. In a study conducted on 1215 patients with DM, the SHR value was shown to be an important parameter in predicting the development of acute renal damage after acute myocardial infarction [20]. In a study conducted by Marenzi et al., the effect of the SHR value on the development of acute renal damage was investigated in DM patients with acute myocardial infarction. A total of 474 patients were included in this prospective study, and acute renal injury developed in 77 (16%) patients. At the end of their studies, the authors determined that the SHR value was a more valuable parameter than the ABG value in predicting renal injury [7]. In a recent study, Liu et al. investigated the importance of the SHR value in predicting acute renal damage after endovascular interventions in patients with ischemic stroke. In this retrospective study, the authors enrolled 717 acute ischemic stroke patients who had undergone endovascular treatment, of whom 205 (28.6%) experienced acute renal injury. At the end of the study, the SHR value was shown to be an independent predictor of the development of acute renal injury (OR: 4.455, 95% CI: 2.237−8.871, *p* < 0.001) [24]. In our study, we obtained the mABG value from the blood glucose values at three different times perioperatively in the EVAR procedure that we planned as elective intervention, and we determined the mSHR value based on this parameter. In the multivariate analysis we conducted in our study, we determined that the mSHR value was an independent predictor of the development of CIN.

Various studies have investigated the risk factors for CIN after endovascular aortic procedures. In one study that included 167 patients with an endovascular stent placed in the thoracic and abdominal aorta, low preoperative left ventricular ejection fraction and increased blood product transfusion were found to be associated with the development of CIN [25]. In another study, a significant relationship was revealed between preoperative creatinine values and elevated contrast agent use, and the development of CIN after endovascular aortic repair [26]. In a study by Guneyli et al., 139 patients were included retrospectively, and risk factors for the development of CIN after endovascular aortic repair were investigated. CIN developed in 39 (28%) patients, and high preoperative serum urea and creatinine levels were identified as important risk factors for the development of CIN. No significant correlation was found between the amount of contrast used in their study and the development of CIN [27]. We retrospectively included 276 patients who had undergone EVAR and found our CIN rate to be 19.9%. We found a significant relationship between increased blood product use, contrast material volume, and preoperative creatinine values, and the development of CIN.

There are some limitations to note in our study. Chief among these is that it was planned retrospectively and that it was a single-center study. As a result, the number of patients was limited. In addition, our study did not include quantitative results of liver fibrosis, such as biopsies. Our results now need to be supported by multicenter prospective studies.

## 5. Conclusions

In recent years, endovascular methods have come to the fore in the treatment of AAAs. This method can be applied to many patients with very low risk rates. However, CIN that can occur after these procedures is an important postoperative complication. In this study, we demonstrated that we may predict the risk of CIN development with preoperatively calculated, noninvasive liver fibrosis scores and mSHR. In the perioperative period, protective measures may be taken by considering these factors.

## Figures and Tables

**Figure 1 healthcare-11-00866-f001:**
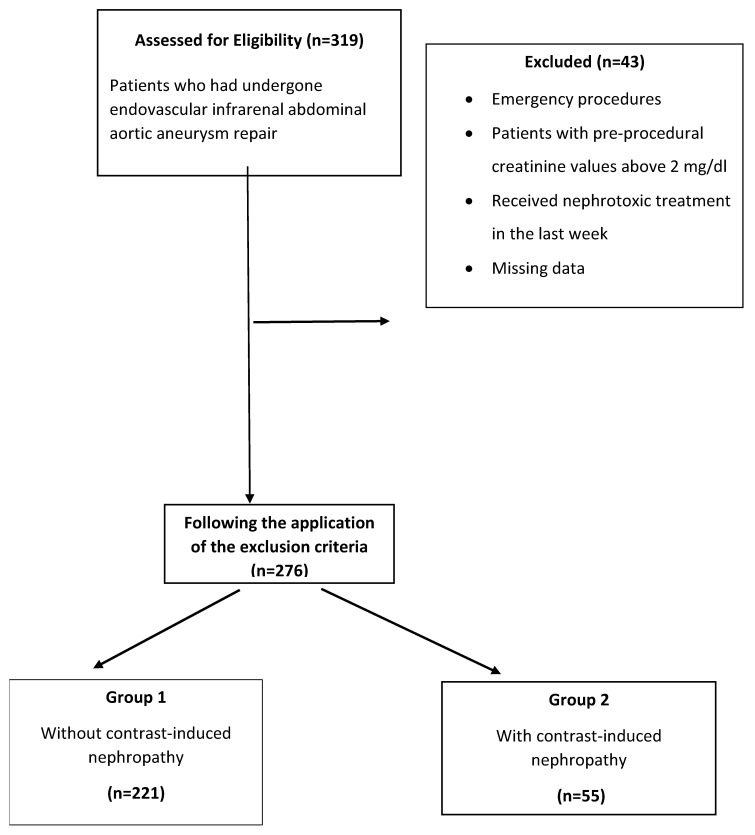
Flow chart of the study.

**Figure 2 healthcare-11-00866-f002:**
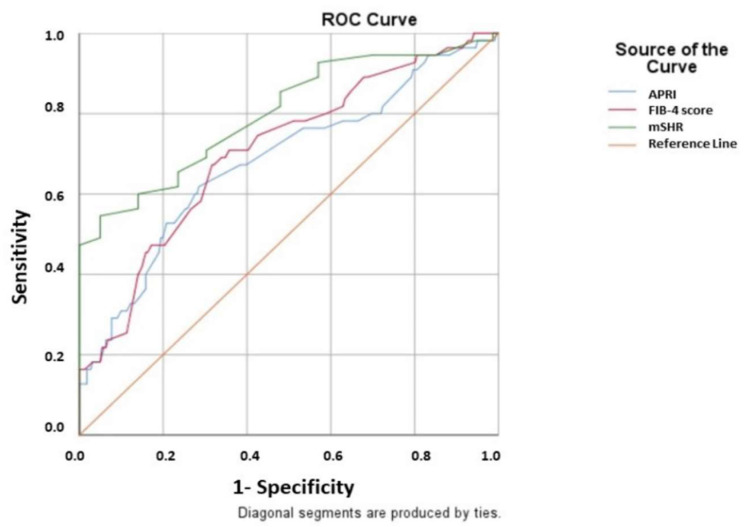
Data figure of the area under the curve (AUC), confidence interval (CI), and cut-off values in receiver operating characteristic (ROC) curve analysis for variables to predict contrast-induced kidney injury (APRI; cut-off: 0.28, AUC: 0.681, 95% CI: 0.596–0.766, *p* < 0.001, with 61.8% sensitivity and 66.7% specificity, (FIB-4 score; cut-off: 1.52, AUC: 0.707, 95% CI: 0.629–0.785, *p* < 0.001, with 69.1% sensitivity and 64.8% specificity), and (mSHR; cut-off: 1.49, AUC: 0.801, 95% CI: 0.727–0.874, *p* < 0.001, with 70.9% sensitivity and 68.3% specificity).

**Table 1 healthcare-11-00866-t001:** Demographic data and preoperative features of the patients.

Variables	Group 1 (N = 221)	Group 2 (N = 55)	*p*-Value
Age (years)	71 (65–88)	74 (65–82)	0.158
Female gender, n (%)	41 (18.6%)	7 (12.7%)	0.308
BMI (kg/m^2^)	26.7 (23.4–33.6)	25.8 (23.2–35)	0.276
Hypertension, n (%)	98 (44.3%)	27 (49.1%)	0.527
Diabetes mellitus, n (%)	40 (18.1%)	14 (25.5%)	0.219
COPD, n (%)	24 (10.9%)	7 (12.7%)	0.695
Smoking, n (%)	38 (17.2%)	10 (18.2%)	0.863
Hyperlipidemia, n (%)	44 (19.9%)	13 (23.6%)	0.541
Ejection fraction (%)	45 (25–65)	40 (25–65%)	0.635
ASA use, n (%)	117 (52.9%)	25 (45.5%)	0.320
ACEI/ ARB use, n (%)	91 (41.2%)	20 (36.4%)	0.515
Calcium channel blocker use n (%)	45 (20.4%)	8 (14.5%)	0.327
Statin use, n (%)	94 (42.5%)	25 (45.5%)	0.696

ACEI: angiotensin-converting enzyme inhibitor, ARB: angiotensin receptor blocker, ASA: acetylsalicylic acid, BMI: body mass index, COPD: chronic obstructive pulmonary disease.

**Table 2 healthcare-11-00866-t002:** Preoperative laboratory variables of the patients.

Variables	Group 1 N = 221	Group 2 N = 55	*p*-Value
White blood cell count (10^3^/µL)	8.1 (4.7–13.2)	7.9 (4.9–14.7)	0.341
Hemoglobin (mg/dL)	11.9 (11.2–15.1)	12.1 (10.9–15.7)	0.196
Platelet count (10^3^/µL)	245 (200–365)	251 (192–358)	0.118
Creatinine (mg/dL)	0.9 (0.7–1.97)	1.2 (0.9–1.99)	<0.001
BUN (mg/dL)	16 (10–36)	18 (12–32)	0.275
AST (IU/L)	23 (22–36)	24 (21–35)	0.291
ALT (IU/L)	21 (16–39)	23 (19–26)	0.116
HbA1c (%)	5.7 (5.1–10.2)	5.9 (4.9–9.8)	0.079
mABG (mg/dL)	148 (126–240)	152 (120–236)	0.008
Na (mEq/L)	137.2 ± 5.5	136.9 ± 6.1	0.334
K (mEq/L)	4.1 (3.6–5.1)	4.2 (3.5–5.3)	0.259
Ca (mg/dL)	9.3 (8.8–10.1)	9.2 (8.7–9.9)	0.694
Mg (mg/dL)	1.91 ± 0.2	1.92 ± 0.2	0.712
CRP (mg/dL)	7.1 (2.4–26.4)	7.5 (2.1–22.3)	0.291
FIB-4 score	1.4 (0.9–1.66)	1.9 (1.12–2.56)	<0.001
APRI	0.22 (0.17–0.33)	0.34 (0.22–0.49)	<0.001
mSHR	0.98 (0.88–1.71)	1.67 (1.23–2.5)	<0.001

BUN: blood urea nitrogen, AST: aspartate aminotransferase, ALT: alanine aminotransferase, HbA1c: hemoglobin A1c, mABG: modified admission blood glucose, Na: sodium, K: potassium, Mg: magnesium, CRP: C-reactive protein, FIB-4 score: fibrosis-4 score, APRI: aspartate transaminase to platelet ratio index, mSHR: modified stress hyperglycemia ratio.

**Table 3 healthcare-11-00866-t003:** Operative and postoperative features of the patients.

Variables	Group 1	Group 2	*p*-Value
	N = 221	N = 55	
Operation time (minutes)	140 (90–240)	144 (90–270)	0.210
Contrast volume (mL)	190 (150–240)	200 (140–250)	0.003
Packed blood products (units)	1 (1–4)	2 (1–5)	0.012
Type of anesthesia			0.289
General (n)	18	7	
Local with sedation (n)	203	48	
Total ICU stay (days)	1 (1–8)	1 (1–10)	0.571
Total hospital stay (days)	5 (4–12)	5 (4–15)	0.391
In-hospital mortality n (%)	3 (1.3%)	1 (1.8%)	1.000

ICU: intensive care unit.

**Table 4 healthcare-11-00866-t004:** Multivariate logistic regression analysis models were used to identify factors affecting postoperative contrast-induced kidney injury.

	Model 1			Model 2		
Variables	*p*-Value	Exp (B) Odds Ratio	95% C.I. Lower–Upper	*p*-Value	Exp (B) Odds Ratio	95% C.I. Lower–Upper
Age	0.361	1.194	0.890–1.655	--	--	--
FIB-4 score	--	--	--	0.005	1.124	1.060–1.592
APRI	0.014	0.791	0.495–0.894	--	--	--
mSHR	<0.001	1.462	1.110–2.125	0.002	1.639	1.350–2.780
Pre-creatinine	<0.001	3.935	2.571–5.150	<0.001	4.120	2.872–6.190
Contrast volume	0.037	0.632	0.398–0.814	0.029	0.590	0.386–0.790
Packed blood products (units)	0.192	1.140	0.890–1.350	0.089	1.075	0.940–1.264

FIB-4 score: fibrosis-4 score, APRI: aspartate transaminase to platelet ratio index, mSHR: modified stress hyperglycemia ratio.

## Data Availability

All data produced here are available and can be produced upon request.

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
