# Peer review of "The Roles of Liver Fibrosis Scores and Modified Stress Hyperglycemia Ratio Values in Predicting Contrast-Induced Nephropathy after Elective Endovascular Infrarenal Abdominal Aortic Aneurysm Repair"

_healthcare, 2023, doi:10.3390/healthcare11060866_

Round 1

Reviewer 1 Report

The authors present a nicely driven study, which tried to assess the ability of different scoring systems in predicting contrast-enhanced nephropathy. The retrospective character of the study is a limitation of the study and substantial revision is required to improve the manuscript, by taking into consideration the following comments:

1. I do not see the need to report the median age of the two groups within the abstract. You can just state that the two study groups were similar in terms of ages.

2. Lines 27-28, abstract: "In Group 2, the contrast volume and packed blood products used were significantly higher." Do you refer to number of packed blood  products used?

3. Introduction: the connection between Fib-4 score, APRI and abdominal aortic aneurysms is not mentioned. There is no data regarding a previously established link between these values and abdominal aortic aneurysms nor contrast induced nephropathy. In line 48 you claim that "Studies have shown their relationship with cardiovascular diseases". What type of cardiovascular diseases do you refer to?

4. Figure 1 has not been cited among the manuscript.

5. It is unclear how mSHR calculation formula was established.

6. Line 94: could you justify through a reference the heparin dosage?

7. Line 103: is this a software?

8. Lines 110-111: "Values with a p-value of <0.2 in univariate analyzes were used in multivariate analysis models." Please clarify why this approach was chosen. You refer to variables in which comparisons of mean values revealed p<0.2? Based on what statistical analysis were this p values obtained?

9. Please clarify within the text what model 1 and model 2 represented.

10. Figure 2 results and what they represent should be detailed more among the result section.

11. English language requires substantial revision and corrections.

12. Informed consent: did the patients agree prior to their procedures to publication of their medical data as well?

Author Response

Reviewer 1

Q1. I do not see the need to report the median age of the two groups within the abstract. You can just state that the two study groups were similar in terms of ages.

A1: Age information of patient groups has been removed from abstract

Q2. Lines 27-28, abstract: "In Group 2, the contrast volume and packed blood products used were significantly higher." Do you refer to number of packed blood  products used?

A2: Number of packed blood products addet to abstract.

Q3. Introduction: the connection between Fib-4 score, APRI and abdominal aorticaneurysms is not mentioned. There is no data regarding a previously established link between these values and abdominal aorticaneurysms nor contrast induced nephropathy. In line 48 you claim that "Studies have shown their relationship with cardiovascular diseases". What type of cardiovascular diseases do you refer to?

A3: In this current study, we aimed to investigate the role of fibrosis 4 score (FIB-4), aspartate transaminase platelet ratio index (APRI) value, and modified stress hyperglycemia ratio (mSHR) values in predicting contrast induced nephropathy (CIN). EVAR is a process with contrast use. In the literature, the relationship between hepatic fibrosis scores with CIN is shown in Patients Undergoing Elective Percutaneous Coronary Intervention (He HM, He C, You ZB, Zhang SC, Lin XQ, Luo MQ, Lin MQ, Zhang LW, Lin KY, Guo YS. Non-Invasive Liver Fibrosis Scores Are Associated With Contrast-Associated Acute Kidney Injury in Patients Undergoing Elective Percutaneous Coronary Intervention. Angiology. 2022 May 31:33197221105745. doi: 10.1177/00033197221105745. ---Reference 13) Also in a study it was revealed that FIB-4 index is associated with an increased incidence of renal failure in the general population------https://doi.org/10.1002/hep4.2104---Reference 10. We added a sentence about CIN and liver fibrosis.

Q4. Figure 1 has not been cited among the manuscript.

A4: Figure 1 was cited

Q5. It is unclear how mSHR calculation Formula was established.

A5: It is revised.SHR= Admission blood glucose level (ABG) (mg/dl)/ [(28.7 × glycosylated hemoglobin %) - 46.7].

Acute hyperglycemia causes prothrombotic events, it increases oxidative stress and leads to endothelial dysfunction. This situation affects the development of adverse events after acute cardiovascular events, regardless of whether the person has diabetes mellitus. In this way, SHR value has recently been shown as a prognostic marker in acute cardiovascular events. Our study determined the ABG value as the average of the blood glucose values taken at 3 different times during the EVAR procedure. We defined the SHR value we calculated with this value as mSHR.

Q6. Line 94: could you justify through a reference the heparin dosage?

A6: We added a reference (9)

Q7. Line 103: is this a software?

A7: Yes it is. Licensed program we use in our hospital

Q8. Lines 110-111: "Values with a p-value of <0.2 in univariate analyzes were used in multivariate analysis models." Please clarify why this approach was chosen. You refer to variables in which comparisons of mean values revealed p<0.2? Based on what statistical analysis were this p values obtained?

A8: In our study, we first performed univariate analysis. In these analyses, we included values close to significance in the multivariate analysis. We performed multivariate analysis in two models. Model 1 has APRI value, age and other variables, while Model 2 has FIB-4 score and other variables. Because of the similar parameters in the calculation of these two scores, two different models were evaluated. Age was not included in the model with FIB-4 score, as the FIB-4 score also included the age variable.

Q9. Please clarify within the text what model 1 and model 2 represented.

A9: We have explicitly written the data in the multivariate analysis models.

Q10. Figure 2 results and what they represent should be detailed more among the result section.

A10: We presented these results wit detail.

Q11. English language requires substantial revision and corrections.

A11: We edited the article

Q12. Informed consent: did the patients agree prior to their procedures to publication of their medical data as well?

A12: We serve as a training and research hospital. Consent for these procedures is obtained from all patients who are operated in our clinic. These consents also contain permissions to use the data. We can use all the data by obtaining ethical committee approval from our clinic without describing a person. In the publications where we describe a patient as a case report, we obtain special consent for this. Therefore, our article does not contain an ethical problem.

All changes made to the article are highlighted in yellow.

Thank you for your patience and understanding

Sincerely yours.

Reviewer 2 Report

The authors address an important issue in endovascular aortic repair - contrast agent induced kidney disease. It is an adequate retrospective approach to study different scores and their relation to postoperative creatinine changes. However, several aspects have to be improved!

Major points:

- line 29 and 244: it is not possible to conclude from a retrospective study that the authors are able to predict the risk for CIN. For this conclusion a prospective study would be mandatory. Although the 

- What is the pathomechanistic link between CIN and liver fibrosis scores?

- line 63: what does 'missing planned data' mean?

- preoperative creatinine values were also significantly higher in group 2 - this fact is not mentioned and discussed

- the models 1 and 2 for multivariate logistic regression are not explained (why, what's the difference?) - this has to be implemented in methods

- the discussion lacks to address potential clinical applications of the gained results. What role do the results have for planing aortic repair, perioperative procedures to protect kidney function or postoperative surveillance? What about prospective data acquisition? Could a score affect patient selection in future?

Minor points:

- figure 1 should be optically improved (straight lines, higher resolution)

- table 3: 'n=221' has to be repositioned

- line 180: 'CAD' is not explained before

- the conclusion seems to be a repetition of the beginning of the introduction, discussion and the abstract and should be revised

- please revise the whole manuscript regarding english language by a native speaker and check for missing our double spaces 

Author Response

Reviewer 2

Major points:

Q1: line 29 and 244: it is not possible to conclude from a retrospective study that the authors are able to predict the risk for CIN. For this conclusion, a prospective study would be mandatory. Although the 

A1: We edited the conclusion. In this current study, we demonstrated that we may predict the risk of CIN development with preoperatively calculated, non-invasive liver fibrosis scores and mSHR. In the peri-operative period, protective measures may be taken by considering these factors.

Q2: What is the pathomechanistic link between CIN and liver fibrosis scores?

A2: CIN involves multiple factors that lead to hypoxia of the medulla and subsequently acute tubular necrosis. . In addition to decreasing activity of nitric oxide (NO) synthase, contrast media inhibit mitochondrial enzyme activities and subsequently increase adenosine through hydrolysis of ATP. Catabolism of adenosine generates reactive oxygen species that scavenge NO. Released along with endothelin and prostaglandin from endothelial cells exposed to contrast media. Moreover, the ectopic lipid is associated with structural and functional changes of proximal tubular cells, podocytes, and mesangial cells. Increasing evidence suggests that intracellular lipid deposition leads to proximal tubule cell dysfunction in the presence of proteinuria (Reference: Nishi H, Higashihara T, Inagi R (2019) Lipotoxicity in kidney, heart, and skeletal muscle dysfunction. Nutrients 11(7))

In the literature, the relationship between hepatic fibrosis scores with CIN is shown in Patients Undergoing Elective Percutaneous Coronary Intervention (He HM, He C, You ZB, Zhang SC, Lin XQ, Luo MQ, Lin MQ, Zhang LW, Lin KY, Guo YS. Non-Invasive Liver Fibrosis Scores Are Associated With Contrast-Associated Acute Kidney Injury in Patients Undergoing Elective Percutaneous Coronary Intervention. Angiology. 2022 May 31:33197221105745. doi: 10.1177/00033197221105745. ---Reference 13) Also in a study it was revealed that FIB-4 index is associated with an increased incidence of renal failure in the general population------https://doi.org/10.1002/hep4.2104. We added a sentence about CIN and liver fibrosis.

Q3: line 63: what does 'missing planned data' mean?

A3: patients with mising data’’  Edited

Q4: preoperative creatinine values were also significantly higher in group2 –this fact is not mentioned and discussed

A4: We discussed it. Preexisting renal insufficiency is the most important risk factor for the development of CIN: The incidence of CIN was 5.3% in 3232 patients with normal renal function and 15.7% in 959 patients with baseline serum creatinine concentration N1.2 mg/dL [Nash K, Hafeez A, Hou S. Hospital-acquired renal insufficiency. Am J Kidney Dis 2002;39:930–6.]

Q5: the models 1 and 2 for multivariate logistic regression are not explained (why, what's the difference?) - this has to be implemented in methods

A5: We edited Methods section. We performed multivariate analysis in two models. Model 1 has APRI value, age and other variables, while Model 2 has FIB-4 score and other variables. Because of the similar parameters in the calculation of these two scores, two different models were evaluated. Age was not included in the model with FIB-4 score, as the FIB-4 score also included the age variable.

Q6: the discussion lacks to address potential clinical applications of the gained results. What role do the results have for planing aortic repair, perioperative procedures to protect kidney function or postoperative surveillance? What about prospective data acquisition? Could a score affect patient selection in future?

A6: In this current study, we demonstrated that we may predict the risk of CIN development with preoperatively calculated, non-invasive liver fibrosis scores and mSHR. In the peri-operative period, protective measures may be taken by considering these factors. Our results now need to be supported by multicenter prospective studies.

 Minor points:

Q1: figure 1 should be optically improved (straightlines, higher resolution)

A1: Figure 1 added as a word format

Q2: table 3: 'n=221' has to be repositioned

A2: Edited.

Q3: line 180: 'CAD' is not explained before

A3: coronary artery disease explained

Q4: the conclusion seems to be a repetition of the beginning of the introduction, discussion and the abstract and should be revised

A4: Conclusion section was edited.

Q5: please revise the whole manuscript regarding english language by a native speaker and check for missing our double spaces 

A5: We edited the whole article regarding English language and writing errors.

All changes made to the article are highlighted in yellow.

Thank you for your patience and understanding

Sincerely yours.

Round 2

Reviewer 1 Report

The authors have addressed my comments and revised the manuscript accordingly.